# The isolated voltage sensing domain of the Shaker potassium channel forms a voltage-gated cation channel

**Juan Zhao[1,2], Rikard Blunck[1,2]\***

[1]Department of Physics, Université de Montréal, Montréal, Canada; [2]Department of Pharmacology and Physiology, Université de Montréal, Montréal, Canada

**Abstract** Domains in macromolecular complexes are often considered structurally and functionally conserved while energetically coupled to each other. In the modular voltage-gated ion channels the central ion-conducting pore is surrounded by four voltage sensing domains (VSDs). Here, the energetic coupling is mediated by interactions between the S4-S5 linker, covalently linking the domains, and the proximal C-terminus. In order to characterize the intrinsic gating of the voltage sensing domain in the absence of the pore domain, the Shaker Kv channel was truncated after the fourth transmembrane helix S4 (Shaker-iVSD). Shaker-iVSD showed significantly altered gating kinetics and formed a cation-selective ion channel with a strong preference for protons. Ion conduction in Shaker-iVSD developed despite identical primary sequence, indicating an allosteric influence of the pore domain. Shaker-iVSD also displays pronounced 'relaxation'. Closing of the pore correlates with entry into relaxation suggesting that the two processes are energetically related.

*For correspondence: rikard. blunck@umontreal.ca

**Competing interests:** The authors declare that no competing interests exist.

## Introduction

Voltage-gated ion channels play a fundamental role in a wide range of physiological processes, including the generation and propagation of electrical signals in excitable cells and muscle tissues, neurotransmitter release in pre-synaptic nerve endings, and maintaining cell homeostasis (*Hille, 1978*). They comprise four subunits of each six transmembrane segments (S1–S6), which are assembled into two functionally distinct parts: a central ion-selective pore formed by the combined S5 and S6 segments of four domains, surrounded by four voltage-sensing domains (VSDs) formed by the four helices S1–S4. S4 is considered the principal component of voltage sensing because it contains positively charged residues that are periodically aligned at every third position. The first four arginines (R1-R4) of S4 sense and interact with the surrounding electric field, and move the S4 outward during depolarization and inward during hyperpolarization in a combination of translation, rotation and tilt (*Faure et al., 2012*; *Li et al., 2014a, 2014b*; *Vargas et al., 2012*; *Yarov-Yarovoy et al., 2012*). These changes are mechanically transmitted to the pore domain via the S4–S5 linker and the C-terminal S6 of the same and neighboring subunit, leading to gating of the central pore (*Batulan et al., 2010*; *Wall-Lacelle et al., 2011*; *Haddad and Blunck, 2011*; *Muroi et al., 2010*; *Long et al., 2005b*; *Catterall, 2010*; *Bezanilla, 2005*; *Lu et al., 2001, 2002*; *Labro et al., 2008*). Opening of the pore has been shown to occur in at least two steps (*Del Camino et al., 2005*; *Kalstrup and Blunck, 2013*). In addition to the coupling in this region, a direct structural and functional interaction between the VSD and pore has been demonstrated (*Lee et al., 2009a*; *Li-Smerin et al., 2000*; *Petitjean et al., 2015*).

The topology and structure of the Kv channels suggest that the VSD and pore domain make up modules that can be adapted to different scenarios (*Blunck and Batulan, 2012*; *Kim et al., 2014*).

**eLife digest** Cells in the heart and other muscles rely on electrical signals to coordinate their activity. They generate these electrical signals by controlling the movement of ions across the membrane that surrounds each cell. Proteins called ion channels in this membrane form pores that allow particular types of ions to pass through. The opening and closing of the pores is tightly controlled so that electrical signals are only generated at specific times.

The Shaker Kv channel is an ion channel that allows potassium ions to pass through the membrane. This protein is made of several modules including one called the voltage sensing domain, which senses changes in the electrical voltage across the membrane to open or close the pore module. However, it is not clear how voltage sensing domain and pore influence each other's structure and behaviour. To address this question, Zhao and Blunck investigated how the voltage sensing domain behaves in animal cells when the pore module is absent.

Zhao and Blunck show that, in the absence of the pore module, the voltage sensing domain becomes an ion channel that allows protons and other positively-charged ions to pass through the membrane. Further experiments reveal this new channel opens at a voltage when the main Shaker Kv channel is usually closed. This may result in small leaks of ions across the membrane that cause long-lasting changes in the timing, intensity and number of electrical signals in cells and organs. The findings of Zhao and Blunck suggest that different modules of ion channels influence each other more than previously thought.

The next steps following on from this work are to explore how the voltage sensing domain channels open and close in more detail and to examine how the pore module influences this behaviour. Investigating whether similar leaks occur in the voltage sensing domains of other ion channels may aid research into some inherited heart or neurological disorders where the channels are cut short between the voltage sensing domain and the pore module.

Chimeras of VSD of one and pore domain of another protein were able to gate in a voltage-dependent fashion (*Lu et al., 2001*; *Arrigoni et al., 2013*). The notion that the VSD acts as an independent module is further supported by proteins where the VSD is linked not to a pore domain but instead to a cytosolic enzymatic domain such as the voltage-sensitive phosphatase of *Ciona intestinalis* (Ci-VSP). It was possible to carry out structural studies on isolated voltage sensor domains (*Li et al., 2014a*; *Chakrapani et al., 2008*; *Jiang et al., 2003*), and the VSD of Ci-VSP has been shown to function in the absence of the phosphatase (*Labro et al., 2012*; *Murata et al., 2005*). Finally, the voltage-gated proton channels (Hv) lack a separate pore domain altogether such that its VSD is directly responsible for proton permeation (*Lee et al., 2009b*; *Li et al., 2015*; *Ramsey et al., 2006*; *Sasaki et al., 2006*). In view of the modular nature of voltage-gated ion channels, VSDs have been considered in structure function studies largely as their own entity with energetic coupling to the pore domain, and they have been directly compared to other VSD-containing proteins such as Ci-VSP and Hv channels. We do not know, however, how possible restraint forces from the pore domain may have obscured conformational changes during VSD activation.

Prolonged depolarization has been shown to reconfigure VSDs of numerous voltage-gated proteins into a stable 'relaxed' state, resulting in a hyperpolarizing shift of the voltage dependence of both pore closure and return of the VSD to its resting position (*Haddad and Blunck, 2011*; *Piper et al., 2003*; *Tan et al., 2012*; *Olcese et al., 1997*; *Kuzmenkin et al., 2004*; *Bruening-Wright and Larsson, 2007*; *Villalba-Galea et al., 2008*). Relaxation or mode shift has been shown to be influenced by the state of the pore domain; it was observed to be correlated with slow C-type inactivation (*Olcese et al., 1997*; *Cuello et al., 2010*; *Männikkö et al., 2005*) and pore stabilization in the open state (*Haddad and Blunck, 2011*). On the other hand, mode shift also developed in Ci-VSP, which does not possess any pore domain, suggesting relaxation to be an intrinsic property of the VSD (*Labro et al., 2012*; *Villalba-Galea et al., 2008*). Recent studies have suggested that the N-terminal tail (*Tan et al., 2012*) and S3-S4 linker length (*Priest et al., 2013*) may affect the VSD relaxation in hERG and Shaker channels, respectively. However, the conformational changes related to relaxation remain unknown.

To gain insights into the gating mechanisms of the Shaker potassium channel VSD as an independent structural and functional unit, we generated a Shaker channel pore deletion mutant, the Shaker isolated voltage sensor domain (Shaker-iVSD). In Shaker-iVSD any conformational changes would not be affected any longer by the energetic load or by structural constraints imposed by the pore domain. Here, we report, for the first time, the gating currents and conformational changes monitored via fluorescence of Shaker-iVSD.

## Results

### Shaker-iVSD forms ion conducting pore

In the pore-deletion mutant Shaker-iVSD, the pore domain (S5-S6) after position I384 and almost the complete C-terminus were removed and a cysteine at position A359 in the S3-S4 linker was introduced to follow conformational changes (*Figure 1a*, *bottom*, for details see Materials and methods). Shaker-iVSD mutants were transiently expressed in *Xenopus laevis* oocytes, and their electrophysiological properties examined using the cut-open oocyte voltage-clamp technique (*Batulan et al., 2010*; *Taglialatela et al., 1992*). Because this deletion mutant does no longer contain the ion conducting pore domain, we used gating current measurements and voltage-clamp fluorometry as an indicator for trafficking to the plasma membrane and responsiveness to changes in membrane potential. Shaker-iVSD was site-directed fluorescently labeled by attaching an extrinsic fluorescent probe (tetramethyl-rhodamine maleimide, TMRM) to a cysteine introduced at position A359C in the S3–S4 linker just N-terminal to S4. As a reference, we used Shaker-A359C-W434F (Shaker-W434F), in which a mutation rendering the channel non-conducting (W434F) and A359C were introduced into background Shaker H4-IR (*Figure 1a*).

Gating currents arising from Shaker-iVSD were detected 4–5 days after cRNA injection using the cut-open oocyte technique (*Figure 1b*, *top*). Our initial objective was to obtain the gating charge-voltage ($QV$) relation of Shaker-iVSD. However, unlike those observed in Shaker-W434F (*Figure 1b bottom*), the gating currents of Shaker-iVSD could not be estimated reliably because of substantial ionic currents in particular when holding the potential at −90 mV. The observed ionic currents in response to a series of 250 ms test pulses applied in 10 mV steps exhibited a weak voltage dependence (*Figure 1c*). In contrast, at the depolarized holding potential of 0 mV, the currents exhibited strong rectification with inward currents at hyperpolarized potentials. The phenotype was consistent with an ion channel, conducting at negative membrane potentials and very slow activation and deactivation kinetics (*Figure 1c and f*). We confirmed that no similar currents developed for Shaker-W434F at similar expression level (*Figure 1d*) and that endogenous currents were negligible in $H_2O$-injected oocytes (*Figure 1f*).

In order to further verify that the currents are not caused by Shaker-iVSD assembling with or inducing expression of endogenous channels in Xenopus oocytes, we expressed Shaker-iVSD in HEK293 cells. The resulting currents in whole-cell patch-clamp recordings show the same ionic currents and voltage dependence as observed in Xenopus oocytes (*Figure 1e*). Holding the membrane potential polarized (−90 mV) led, as in Xenopus oocytes, to constitutively open channels, whereas holding at 0 mV led to hyperpolarization-activated ionic currents with slightly less rectification than in Xenopus oocytes (*Figure 1g*).

### Selectivity of Shaker-iVSD pore

Since the inward ionic current was also observed with $NMDG^+$ (N-methyl-D-glucamine) as the dominant ion in the extracellular solution, we tested the proton selectivity of the Shaker-iVSD-induced current. The proton driving force was altered by shifting the proton reversal potential, $E(H^+)$, through alterations of the transmembrane pH gradient. Oocytes were bathed in 70 mM $NMDG^+$ solutions with extracellular pH modulated to 9.5, 7.5 and 4.5 and intracellular pH maintained at 7.5. Consistent with a proton selective pore, the inward currents increased and E(H+) shifted to more positive potentials when decreasing external pH, although the reversal potential did not reach $E(H^+)$ completely (12.5 ± 1.6 mV for pH 4.5 and −5.4 ± 1.8 mV for pH 9.5; *Figure 2a*).

The discrepancy between $V_{rev}$ and $E(H^+)$ may be due to the imperfect control of the local proton concentration near the membrane, or due to permeation of other ions through the channel. To distinguish between these two possibilities, we measured the reversal potential $V_{rev}$ after reducing the

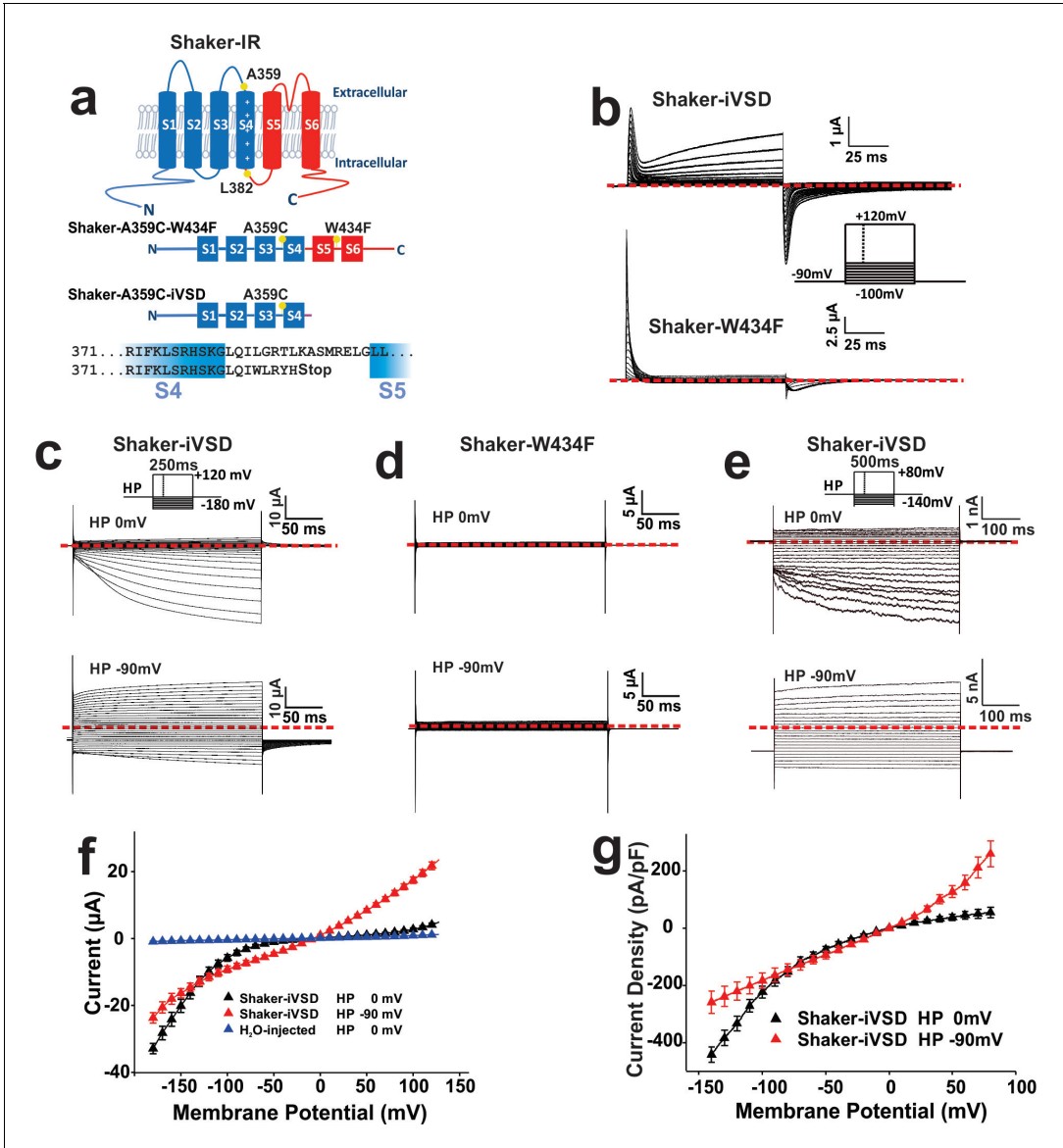

**Figure 1.** Expression of Shaker-iVSD lacking pore domains is sufficient to reconstitute voltage-dependent channel activity. (a) *Top*: topology of Shaker K$^+$ channel. Amino acid positions relevant to the present study are highlighted in yellow. *Middle*: Shaker channels constructs (Shaker-W434F and Shaker-iVSD) used in the present study. Thiol-reactive fluorophores are attached to a cysteine at position A359C in the S3–S4 linker. The mutation W434F in the pore region produces an instantly C-type inactivated Shaker channel. Shaker-iVSD is a deletion mutant truncated after the S4 helix. *Bottom:* Sequence of the C-terminus of Shaker-iVSD (b) Gating currents recorded from oocytes expressing Shaker-iVSD (*top*) and Shaker-W434F (*bottom*) using cut-open oocyte voltage clamp. Leak, background, and capacitive currents were subtracted from the current traces using a P/4 protocol. Inset shows the current step protocol. Gating currents were elicited by test pulses from −100 mV to +120 mV at a holding potential of −90 mV. Zero level was indicated by the red dashed line. (c and d) Representative ionic currents recorded by cut-open oocyte voltage-clamp from oocytes expressing Shaker-iVSD and Shaker-W434F. At depolarized (0 mV, *top*) or hyperpolarized (−90 mV, *bottom*) holding potentials, currents were elicited by voltage pluses at a range between −180 and +140 mV in 10 mV increments without leak substraction. The interval between test pulses was 5 s to allow complete recovery of the channels. An inset on *top* of current traces shows the corresponding voltage protocols. Zero level was indicated by the red dashed line. NMDG$^+$-based solutions were used as the external and internal solutions, pHin/pHout 7.35/7.35. (e) Representative ionic currents recorded from HEK293 cells expressing Shaker-iVSD using the whole-cell configuration of the patch-clamp technique. Cells were depolarized for 500 ms to potentials ranging from −140 mV to +100 mV (at a holding potential of 0 mV, *top*), or from −140 mV to +80 mV (at a holding potential of −90 mV, *bottom*) in 10 mV increments. An inset on *top* of current traces shows the corresponding voltage protocols. Zero level was indicated by the red dashed line. To compare with the ionic current recorded from oocytes as shown in c, NMDG$^+$-based solutions were also used as the extracellular and intracellular solutions, pHin/pHout 7.35/7.35. (f) *I-V* relations of currents recorded from Shaker-iVSD-injected oocytes at holding potentials of 0 mV (black triangle, n = 14) or −90 mV (red triangle, n = 13), and H$_2$O-injected oocytes at holding potentials of 0 mV (blue triangle, n = 5) using the cut-open oocyte voltage clamp technique (see protocol in the upper inset of c). The mean steady-state currents during the last 50 ms of the command

*Figure 1 continued on next page*

Figure 1 continued

pulses were plotted versus voltage. (g) I-V relations of ionic currents recorded from HEK293 cells transfected with Shaker-iVSD at holding potentials of 0 mV (black triangle, n = 8) or −90 mV (red triangle, n = 4) using the whole-cell patch clamp technique (see protocol in the upper inset of e). The mean steady-state currents during the last 50 ms of the command pulses were plotted versus voltage.

ionic strength by D-sorbitol dilution while keeping the pH constant. Reducing the ionic strength should not alter the reversal potential if the current was perfectly proton selective since the proton concentration remains constant while all other ion concentrations were reduced (**DeCoursey, 2013**). Substitution of $NMDG^+$ with D-sorbitol in the external solution to a final concentration of 10 mM (versus 115 mM cytosolic) substantially decreased the normalized inward current and shifted the $V_{rev}$ from $−1 \pm 1$ mV to $−62.5 \pm 2.5$ mV at symmetric pH = 7.35, suggesting that even $NMDG^+$ ions are to a certain extent permeant charge carriers of Shaker-iVSD-induced currents (**Figure 2b**).

A proton gradient of $pH_{in}/pH_{out}$5.5/7.5 and 4.5/7.5 at a low $NMDG^+$ concentration (10 mM) and high buffer concentrations in both external and internal solutions resulted in a $V_{rev}$ $−7.0 \pm 1.1$ mV and $−34.9 \pm 1.8$ mV, respectively (**Figure 2c**). These experiments were performed on whole-cell patches from HEK293 cells. The relative permeability for protons over $NMDG^+$ was $P_H/P_{NMDG}$ ~ 1400 according to the Goldman-Hodgkin-Katz equation. The experiments also excluded any significant permeability for anions (symmetric 10 mM).

To identify the relative selectivity of Shaker-iVSD to different cations, we performed ionic substitution experiments in oocytes using the cut-open oocytes technique. Normalized current amplitudes of Shaker-iVSD recorded in external solutions containing either $Na^+$ or $K^+$ as the predominant extracellular cation were 1.35 and 1.54 times higher than in $NMDG^+$ containing external solution (**Figure 2d–e**). The order of permeation efficiency was $P(H^+) \gg P(K^+) \approx P(Na^+) > P(NMDG^+)$. Replacement of the anion mesylate ($MeSO_3^-$) with chloride in the external solution (NMDG-$MeSO_3$ vs NMDG-Cl) did not alter the Shaker-iVSD-induced current, confirming that the current is not carried by anions (**Figure 2f**).

## Permeation pathway of iVSD

Ionic currents through the VSD of the Shaker channel have been described for protons and other cations upon mutations along the S4 (**Starace and Bezanilla, 2004**; **Moreau et al., 2015**; **Tombola et al., 2005**), and the transmembrane domain of Hv1 proton channels consist of only S1-S4 (**Ramsey et al., 2006**; **Sasaki et al., 2006**; **Takeshita et al., 2014**). We thus expected that the cations follow a similar path through the VSD. Since Hv channels are blocked by $Zn^{2+}$ in the micromolar range (**Ramsey et al., 2006**; **Sasaki et al., 2006**), we tested Shaker-iVSD's sensitivity to the application of external $Zn^{2+}$. We found that with increasing concentrations of $Zn^{2+}$ (0–20 μM), the ionic current of Shaker-iVSD continuously reduced (**Figure 3a and c**).

While we were not able to record gating currents reliably, we were able to trace the conformational changes occurring in Shaker-iVSD using voltage-clamp fluorometry. Attaching TMRM at position A359C reports conformational changes related to the gating currents (**Haddad and Blunck, 2011**). Oocytes, expressing Shaker-iVSD and labeled with TMRM, exhibited clear fluorescence signals two days after injection (**Figure 3b and d**). The time course of the normalized fluorescence change did not alter during activation in the presence of $Zn^{2+}$ but deactivation was slowed (**Figure 3b and d**). Accordingly, also the fluorescence voltage (FV) relation remained unaltered. While the activation kinetics and FV remained unaltered, the relative fluorescence change was diminished with increasing $Zn^{2+}$-concentration similar to the decrease in current amplitude (**Figure 3b and e**). A reduced relative fluorescence change indicates that immobilization of the voltage sensor by $Zn^{2+}$ caused the current decrease. Fluorophores attached to voltage sensors immobilized by $Zn^{2+}$ would still contribute to the total fluorescence intensity (F) but would no longer be displaced and thus not display a voltage dependent change (dF). Accordingly, the relative fluorescence change dF/F would be lower. This was further validated by the reduction in gating currents with addition of $Zn^{2+}$ (**Figure 3—figure supplement 1**).

In Hv1 channels, histidines at position H140 and H193 have been shown to be responsible for the $Zn^{2+}$ sensitivity (**Ramsey et al., 2006**; **Takeshita et al., 2014**). These histidines are positioned at the entry to the gating pore, right above the positively charged arginines of the S4 (**Figure 3f**) and

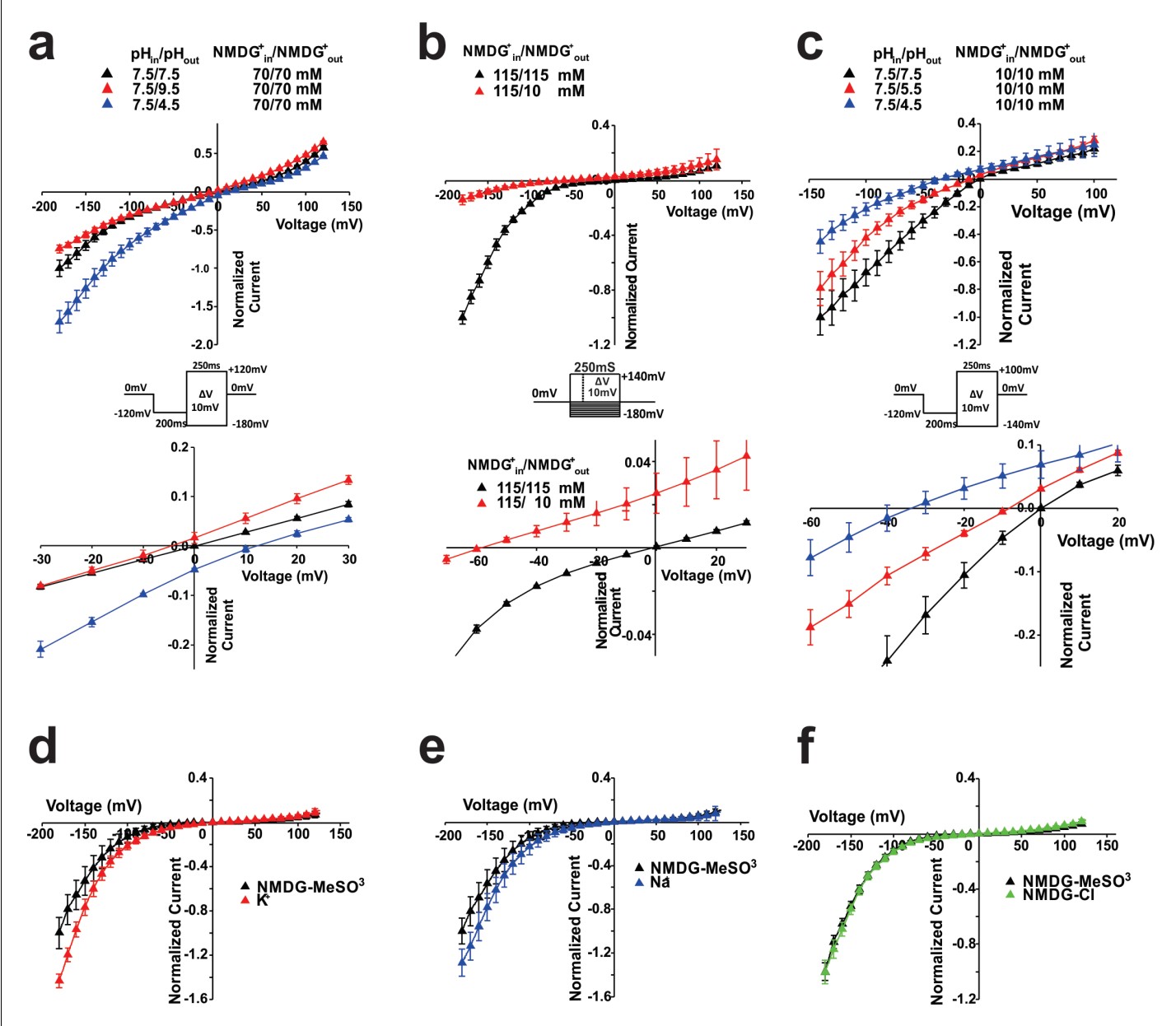

**Figure 2.** Selectivity of Shaker-iVSD-induced current in high buffer (HB) solutions or NMDG⁺-based solutions. (**a**)*Top.* Normalized *IV* relationships of Shaker-iVSD induced currents under different external pH conditions with HB solutions. Tail currents were evoked by 250 ms voltage pulses ranging from −180 to +120 mV following a 200 ms hyperpolarization prepulse to −120 mV using cut-open oocyte technique (see inset for protocol). External HB solution contained (in mM) 70 NMDG⁺, and either 180 acetic acid (for pH 4.5), 180 HEPES (for pH 7.5), or 136 CHES with 44 D-sorbitol (for pH 9.5). The composition of the HB internal solution was the same as the HB internal solution at pH 7.5. *Bottom.* Same normalized *IV* relationships, but zoomed in to emphasize the changes of $V_{rev}$. At pHin/pHout 7.5/7.5, $V_{rev}$ = −0.14 ± 0.18 mV (black triangle, n = 56); at pHin/pHout 7.5/4.5, $V_{rev}$ = 12.5.2 ± 1.6 mV (blue triangle, n = 5); at pHin/pHout 7.5/9.5, $V_{rev}$ = −5.4 ± 1.76 mV (red triangle, n = 5). (**b**) *Top.* Normalized *IV* relations of currents with NMDG⁺in/ NMDG⁺out 115 mM/10 mM or 115 mM/115 mM, at pHin/pHout 7.35/7.35. NMDG⁺-based external and internal solutions contained 115 mM NMDG-MeSO₃ were the same as in *Figure 1b–d*. To test the permeability of Shaker-iVSD-induced currents to NMDG⁺ ions, 115 mM NMDG-MeSO₃ in the external solution was replaced by 10 mM NMDG-MeSO₃ and 210 mM D-sorbitol, at pH 7.35. Currents were elicited by the voltage pulse protocol shown in the inset. *Bottom.* Same normalized *IV* relationships, but zoomed in to emphasize the changes of $V_{rev}$. At NMDG⁺in/NMDG⁺out 115 mM/10 mM, $V_{rev}$ = −62.6 ± 2.5 nmV (red triangle, n = 4); at NMDG⁺in/NMDG⁺out 115 mM/115 mM, $V_{rev}$ = −0.8 ± 1.1 (black triangle, n = 4). (**c**) *Top.* Normalized *IV* relationships of currents recorded from HEK293 cells transfected with Shaker-iVSD cDNA with HB solutions. Tail currents were elicited from a holding potential of 0 mV by 250 ms voltage pulses ranging from −140 to +100 mV in 10 mV increments, following a 200 ms hyperpolarization prepulse to −120 mV using whole-cell patch clamp technique (see inset for protocol). The intracellular and extracellular HB solutions contained (in mM) 10 NMDG⁺, and either 24 HEPES (for PH 7.5), 56 MES (for pH5.5), or 35 acetic acid (for pH4.5). Osmolarity was adjusted to 300~320 mOsm by

*Figure 2 continued on next page*

*Figure 2 continued*

D-sorbitol. *Bottom.* Same normalized *IV* relationships, but zoomed in to emphasize the changes of $V_{rev}$. $V_{rev}$ values were 0.5 ± 0.2 mV at pHin/pHout 7.5/7.5 (black triangle, n = 9), −7.0 ± 1.1 mV at pHin/pHout 5.5/7.5 (red triangle, n = 6), and −34.9 ± 1.8 mV at pHin/pHout 4.5/7.5 (blue triangle, n = 10). (**d–e**) Comparison of conductance of Shaker-iVSD for external $NMDG^+$ (black triangles, n = 12), $K^+$ (*f*, red triangles, n = 7) and $Na^+$ (*g*, blue triangles, n = 7) using cut-open oocyte technique. The $NMDG^+$-based external solution and internal solutions were the same as in *Figure 1b–d* at pHin/pHout 7.35/7.35. $Na^+$-based or $K^+$-based external solutions were the same as the $NMDG^+$-based external solution except that 115 mM NMDG-$MeSO_3$ was replaced by the same concentration of NaOH- $MeSO_3$ or KOH- $MeSO_3$. Currents were elicited by the voltage pulse protocol shown in the inset of *Figure 1c*, and currents were normalized relative to the maximum $NMDG^+$ inward currents and plotted versus voltage. (**f**) Replacing $MeSO_3$ with chloride (green triangle, n = 7) in the external $NMDG^+$-based solution did not exhibit significantly difference from the currents recorded in external NMDG-$MeSO_3$ solution, indicating that it is likely not carried by anions.

would thus prevent proton conduction through the pore. While the histidines are not conserved in Shaker-iVSD, H140 and H193 align with F280 and E335, respectively. F280 unlikely coordinates a $Zn^{2+}$, this role is more likely taken over by either D277 or E283. The lower affinity of glutamate and aspartate might be responsible for the lower sensitivity of Shaker-iVSD to $Zn^{2+}$, as we find ~40% of the current still at 20 µM, at which Hv1 is already fully blocked (*Takeshita et al., 2014*).

Wild type Shaker channels are stabilized in the closed state by external (300 µM) $Zn^{2+}$ (*Boland et al., 1994*). In Shaker-iVSD, this would reflect the conducting state. However, the addition of external $Zn^{2+}$ to Shaker-iVSD did reduce and not increase the current at −90 mV, suggesting that either binding of $Zn^{2+}$ caused the voltage sensing domain to adopt its 'native' conformation (i.e. the non-conducting one in the full Shaker channel) or that the $Zn^{2+}$ directly blocks the ion permeation pathway. The position of D277/E283 and E335 – corresponding to H140 and H194 – support the notion that the ion permeation follows the pathway through the gating pore like in Hv1 and ω-currents.

To corroborate this finding, we substituted D277 with histidine (D277H) and tested the $Zn^{2+}$-sensitivity. (Unfortunately, the double mutant D277H-E335H did not express sufficiently.) The voltage dependence of opening of Shaker-iVSD-D277H was shifted to more hyperpolarized potentials ($V_{1/2}$ = −99 mV). While the kinetics were very similar (*Figure 3g–h*). When blocking the currents with $Zn^{2+}$, we found that the channel showed a 40-times higher affinity to $Zn^{2+}$ with an IC50 of 0.15 µM versus 6 µM for Shaker-iVSD (*Figure 3i*), confirming the pathway through the gating pore in the central voltage sensor domain.

## Kinetics of the iVSD channel

The development of ionic current in the absence of a pore domain shows that the Shaker-iVSD adopts a different conformation than the voltage sensing domain in the complete channel protein. This raises the question as to how the absence of the pore influences the gating kinetics. To analyze the gating kinetics of Shaker-iVSD in detail, we elicited fluorescence traces in response to 250 ms voltage clamp pulses to a range of potentials from −180 mV to +160 mV. The resulting fluorescence kinetics of Shaker-iVSD were slower than Shaker-W434F (*Figure 4a*). At a holding potential of −90 mV, the channels were already in the open state, and they closed (or deactivated) very slowly upon depolarization. The corresponding fluorescence changes did not seem to correlate to the ionic current.

At a holding resting potential of 0 mV, it became more feasible to compare the slower development of the ionic currents (*Figure 1c top*) to the corresponding fluorescence changes (*Figure 4a, center*). The fluorescence developed slower than at −90 mV, and superposition of ionic current and fluorescence changes from simultaneous measurements showed that both current and fluorescence developed with similar time constants when pulsing to strongly hyperpolarized potentials (*Figure 4b*).

The fact that the signals coincided during hyperpolarization (holding at 0 mV) but not during depolarization (holding at −90 mV) shows that fluorescence change and conduction are two independent steps. The step developing conduction is rate limiting only when holding at 0 mV. From a holding potential of −90 mV, the iVSD first undergoes the fast conformational change, reported by the fluorescence that corresponds to the gating currents (*Haddad and Blunck, 2011*), before the channels close slowly. Conversely, when holding at 0 mV, iVSD first opens the channel (slow,

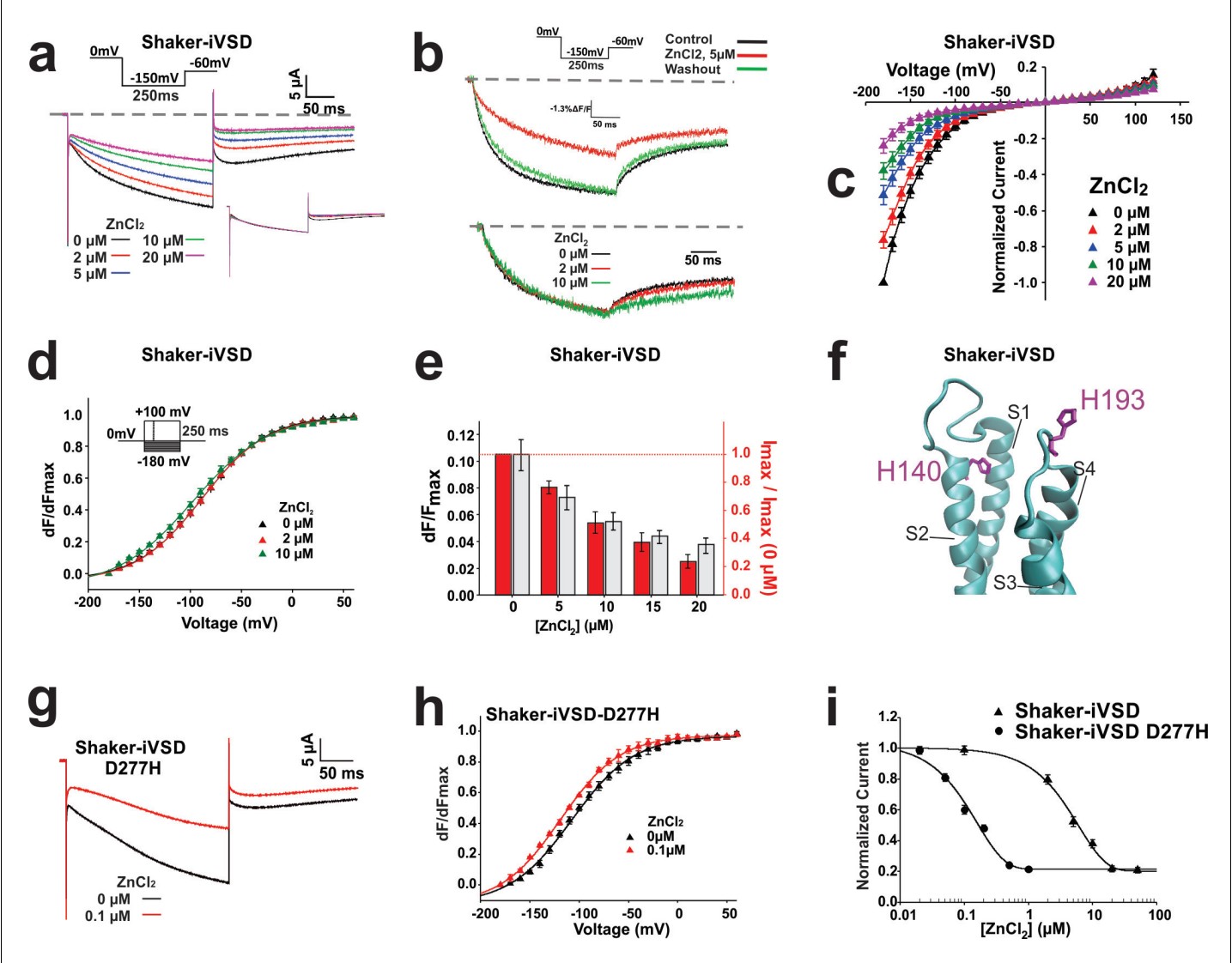

**Figure 3.** Inhibition of Shaker-iVSD-induced ionic currents by external ZnCl2. (a) Ionic currents and normalized fluorescence traces recorded from oocytes expressing Shaker-iVSD before and after the application of 2~20 µM $ZnCl_2$ using cut-open technique (*see inset for protocol*). Zero level is indicated by the grey dashed line. To illustrate the effects of $ZnCl_2$ on kinetics, the same ionic currents were normalized to the value at the end of the 250 ms pulse (*see lower inset*). (b) Fluorescence traces to the recordings in **a** obtained from fluorescent labeling at position A359C atop S4. (c) IV-relations of Shaker-iVSD at increasing concentrations of $ZnCl_2$. (*for protocol see* **a**). Currents were normalized to the maximal current amplitude in the absence of $Zn^{2+}$. (d) Normalized *FV* relations in the absence (*black*) and presence of external 2 µM (*red*) or 10 µM (*green*) $ZnCl_2$. Fluorescence curves were generated using the protocol shown as inset. Smooth curves are fits to a Boltzmann function yielding the following $V_{1/2}$ and dV values: −88.2 ± 1.2 and 32.2 ± 1.7 mV for Shaker-iVSD control (n = 6); −89.6 ± 2.3 mV and 32.8 ± 2.6 mV for 2 µM $ZnCl_2$(n = 6); −98.4 ± 2.9 mV and 37.5 ± 2.0 mV for 10 µM $ZnCl_2$(n = 6). Each curve was normalized to its own maximal relative change dF/F. (e) Effect of $Zn^{2+}$ on relative fluorescence change (dF/F, *grey*) and ionic current. The relative fluorescence change remains constant even during bleaching (*Blunck et al., 2004*), assuming not too high background fluorescence. A reduction of the relative fluorescence change is thus correlated with immobilization of the voltage sensors. The current values are the maximal current amplitudes of **c** normalized to the value in the absence of $Zn^{2+}$. (f) Position of H140 (S2) and H193 (S3-S4 linker; numbering according to human Hv1) in the crystal structure of Hv1 (PDB: 3WKV) in the entry to the gating pore. Missing residues in the crystal structure were modeled using *Modeller* (*Sali and Blundell, 1993*). (g) Ionic current traces recorded from oocytes expressing Shaker-iVSD D277H in the absence (black) and presence (red) 0.1 µM $ZnCl_2$ using cut-open technique (see inset in **a** for protocol). (h) Normalized *FV* relations in the absence (black triangle, n = 5) and presence (red triangle, n = 5) of 0.1 µM $ZnCl_2$. Fluorescence signals were generated using the protocol shown as inset in **d**. Smooth curves are Boltzmann fits to the data with the following $V_{1/2}$ and dV values: −99.0 ± 2.5 mV and 25.9 ± 1.3 for control; −110.4 ± 0.2 mV and 25.7 ± 1.9 mV for $ZnCl_2$. (i) Dose-response curves of inhibition by external $ZnCl_2$ for Shaker-iVSD and Shaker-iVSD D277H ionic currents. Data points correspond to normalized amplitudes of ionic currents elicited by step hyperpolarization to −150 mV (As shown in **a**), plotted as a function of $ZnCl_2$ concentration, and fitted with the Hill equation. The IC50 value of $ZnCl_2$ for the Shaker-iVSD D277H inward currents was 158.4 nM (maximum inhibition 79%, n = 5), which was significantly lower than the IC50 value obtained from Shaker-iVSD currents 6.04 µM (maximum inhibition 80%, n = 6).

*Figure 3 continued on next page*

*Figure 3 continued*

The following figure supplement is available for figure 3:

**Figure supplement 1.** Effect of $Zn^{2+}$ on fluorescence (*left*) and gating currents (*right*).

conduction) before undergoing the normal gating movement (fast, fluorescence). The fluorescence now follows the rate-limiting slow channel opening.

In a more in-depth analysis activation kinetics of ionic current traces (as shown on the top left of *Figure 1c*) and fluorescence signals (as shown on the top of *Figure 4*) followed a bi-exponential function with a fast and a slow time constant ($\tau_{fast}$ and $\tau_{slow}$, respectively, *Figure 4c*). The time constants were voltage-dependent and showed a typical bell shape with maxima at about −100 mV for fluorescence signals and −90 ~ −70 mV for ionic currents. The time constants of current and fluorescence signal correlated well in the range where most of the ionic current developed (<−120 mV) whereas the ionic current developed slower at more depolarized potentials. Although we found the same time constants in current and fluorescence, the time courses did not superimpose directly (*Figure 4b*), indicating that the processes associated to both time constants did not contribute equally to the fluorescence amplitudes.

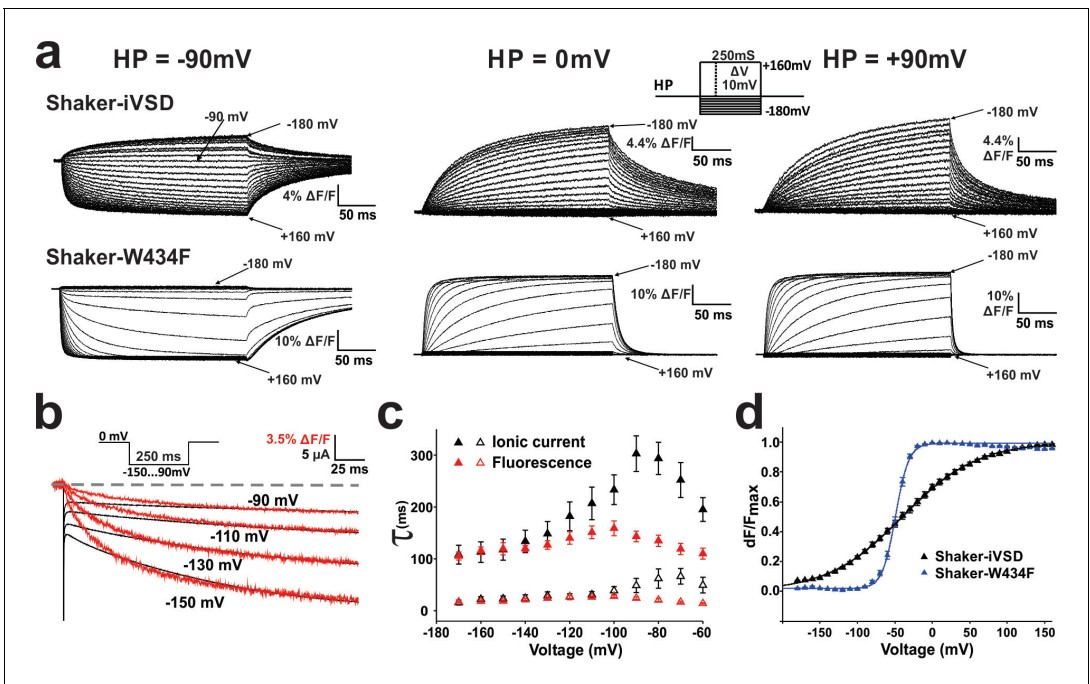

**Figure 4.** Conformational changes in Shaker-iVSD. Shaker-iVSD displays slow voltage-dependent fluorescence quenching in oocytes under voltage-clamp using cut-open oocyte technique. (**a**) Typical fluorescence signals are shown for Shaker-iVSD and Shaker-W434F for 250 ms pulses between −180 and +160 mV from holding potentials of −90 mV (*left*), 0 mV (*center*), and +90 mV (*right*). The inset shows the corresponding clamp protocol. (**b**) Scaled and overlaid representative ionic current (*black*) and fluorescence traces (*red*) in response to a voltage step from 0 mV to −150,−130, −110 and −90 mV for Shaker-iVSD (see inset for protocol). Note that the fluorescence signal has been inverted for a direct kinetic comparison with the current traces. (**c**) The activation kinetics of the Shaker-iVSD ionic currents (as shown on the top of *Figure 1c*) and fluorescence traces during channel activation (as shown in a) were fitted by two exponential components, $I_{fast}$ and $I_{slow}$ for ionic currents (n = 7), $F_{fast}$ and $F_{slow}$ for fluorescence traces (n = 7). The fitted time constants were plotted as a function of test potential. (**d**) FV relationships of Shaker-iVSD and Shaker-W434F at holding potential of −90 mV calculated by normalizing the mean ΔF values during the last 50 ms of the command pulse to the maximum fluorescence change, and plotted against voltage. Smooth curves were fits to a Boltzmann function yielding the following $V_{1/2}$ and *dV* values: −38.8 ± 3.4 and 49.3 ± 2.3 mV for Shaker-iVSD (n = 12), −49.2 ± 1.3 and 8.6 ± 0.4 mV for Shaker-W434F (n = 12).

## iVSD gating and mode shift upon prolonged depolarization

In the fluorescence voltage-relation, the midpoint of activation was not significantly shifted in Shaker-iVSD compared to wild type, but the voltage dependence was much shallower, so much so that even at the extreme potentials the *FV* relation were not yet fully saturated (*Figure 4d*). However, we have shown above that the kinetics of the fluorescence changes differed significantly when holding at −90 mV as compared to holding at 0 mV (*Figure 4a*). The differences also stretched towards the corresponding voltage dependences of the fluorescence voltage relations, which shifted by ~50 mV when holding at both voltages (*Figure 5a*). Such a hysteresis was earlier identified as 'relaxation' or 'mode shift' (*Bruening-Wright and Larsson, 2007*; *Villalba-Galea et al., 2008*). We previously described that the pore domain contributes to the development of the mode shift (*Haddad and Blunck, 2011*) and it had been related to C-type inactivation (*Olcese et al., 1997*). However, mode shift, or 'relaxation', was also observed in the voltage-gated phosphatase of *C. intestinalis* (Ci-VSP), which is devoid of a pore domain (*Labro et al., 2012*; *Villalba-Galea et al., 2008*), suggesting that it is an intrinsic property of voltage sensing domains. Our finding here resolves the issue since also Shaker-iVSD follows the hysteresis pattern.

Still, the conformational changes associated with relaxation remain unknown; we therefore investigated whether, in Shaker-iVSD, the development of an ion conducting pore is related to relaxation. We determined the *FV* relationships for Shaker-iVSD for holding potentials between −90 mV and +90 mV (*Figure 5a*). Just like wild type Shaker channels, the *FV* relations were shifted to more hyperpolarized potentials when holding at depolarized potentials. The shift of the *FV* relations was more pronounced in Shaker-iVSD, shifting by −95 mV compared to −31 mV shift of Shaker-W434F (*Figure 5b*). The *FV* relations of Shaker-iVSD were continuously shifted to more hyperpolarized potentials at more depolarized holding potentials and did not saturate within the tested holding potentials. Unfortunately, holding at more extreme potentials did not yield reliable data since the oocytes did not remain stable for sufficiently long time. In Shaker-W434F, the shift already saturated at a holding potential of −30 mV.

The much more pronounced mode shift in the isolated voltage sensor versus the complete Shaker channel indicated that the Shaker-iVSD requires much stronger hyperpolarization to return to its resting position when lacking the pore domain. We were interested in comparing the time course for entering the mode shift of Shaker-iVSD with the development of the ionic current. We applied pre-pulses of increasing duration from a holding potential of 0 mV to −90 mV, followed by a series of test pulses, determining the relative fluorescence change. For Shaker-iVSD, the resulting curve was mono-exponential with a time constant of 452 ms, whereas recovery from relaxation occurred in double-exponential fashion with time constants of 36 ms and 279 ms for Shaker-W434F (*Figure 5c–e*). The mode shift coincided temporally with the onset of the ionic current in Shaker-iVSD (*Figure 5f*), indicating that the permeation pathway, although open in both resting and activated state, closes during relaxation of the voltage sensing domain.

## Discussion

The voltage sensing domain of the Shaker Kv channel altered its behaviour significantly when expressed in the absence of the pore domain. The voltage dependence is much shallower than the voltage dependence of Shaker-WT and it is evident from the development of cation conduction that the voltage sensing domain assumes a different structural conformation than in the full-length channel. The shallower voltage dependence is probably related to a dispersion of the electric field in the voltage sensor; with the 'hydrophobic plug' or 'gating charge transfer center' (*Tao et al., 2010*) no longer intact, the electric field will be less focussed and thus diminish the effective valence of the gating transition. There are two possible explanations for the rearrangements, (i) the missing interaction with the pore domain leads to a different, energetically more favourable state of the entire voltage sensing domain, or (ii) uncoupling from the pore domain allows the voltage sensor to continue its normal gating movement and enter a deeper resting state not accessible in the wild type. If the second case were true, the deactivation kinetics (i.e. here: return to 0 mV) should remain unaltered, but we observed significant slowing of deactivation (*Figure 4a*). More importantly, activation of the voltage sensor, monitored by the fast fluorescence change, did not close the ion conduction pathway: the activation kinetics when holding at −90 mV (*resting*), measured via fluorescence, were fast (*Figure 4a*), and the current did not reduce during a 250 ms-pulse (*Figure 1c*). iVSD thus leaves the

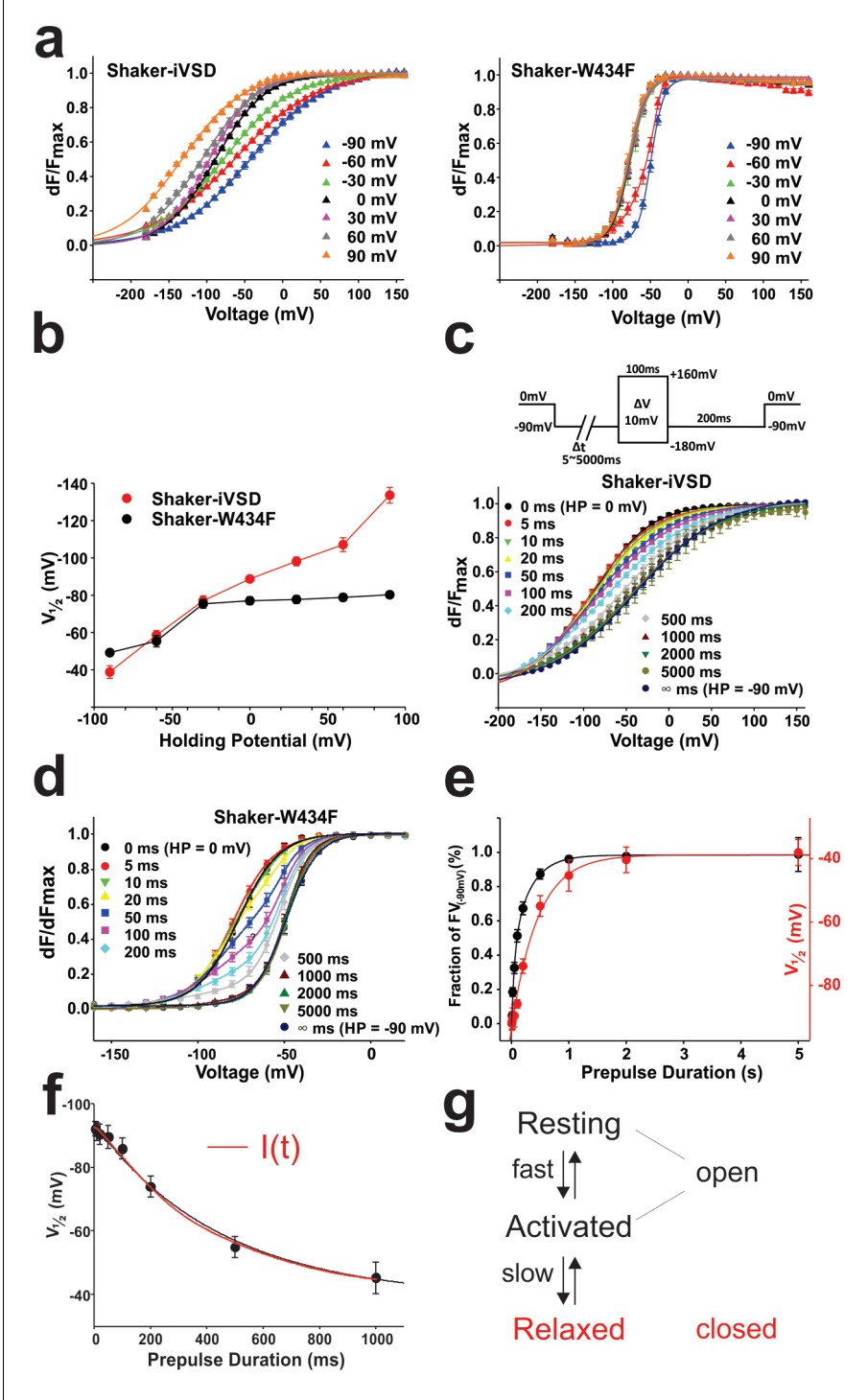

**Figure 5.** Mode shift in Shaker-iVSD. (**a**) *FV* relationships at 7 different holding potentials of Shaker-iVSD (*left*) and Shaker-W434F (*right*). Smooth curves were fits to a Boltzmann function yielding the following $V_{1/2}$ and *dV* values: for Shaker-iVSD, $-38.8 \pm 3.4$ and $49.3 \pm 2.3$ mV at HP $-90$ mV (n = 12), $-58.5 \pm 2.5$ and $57.4 \pm 2.3$ mV at HP $-60$ mV (n = 24), $-77.0 \pm 2.7$ and $44.9 \pm 1.3$ mV at HP $-30$ mV (n = 20), $-88.7 \pm 1.4$ and $33.6 \pm 1.5$ mV at HP 0 mV (n = 24), $-98.2 \pm 2.4$ and $32.0 \pm 2.0$ mV at HP $+ 30$ mV (n = 9), $-107.1 \pm 3.7$ and $37.2 \pm 2.0$ mV at HP $+ 60$ mV (n = 11), $-133.7 \pm 4.2$ and $40.6 \pm 1.9$ mV at HP $+ 90$ mV (n = 7); for Shaker-W434F, $-49.2 \pm 1.3$ and $8.6 \pm 0.4$ mV at HP $-90$ mV (n = 12), $-5.5 \pm 3.0$ and $14.2 \pm 0.9$ mV at HP $-60$ mV (n = 14), $-75.4 \pm 2.4$ and $11.6 \pm 1.1$ mV at HP $-30$ mV (n = 14), $-77.1 \pm 2.0$ and $11.2 \pm 0.9$ mV at HP 0 mV (n = 7), $-77.7 \pm 2.1$ and $11.7 \pm 0.8$ mV at HP $+ 30$ mV (n = 7), $-78.9 \pm 2.0$ and $11.2 \pm 0.8$ mV at HP $+ 60$ mV (n = 8), $-80.3 \pm 1.8$ and $10.8 \pm 0.5$ mV at HP $+ 90$ mV (n = 6).

*Figure 5 continued on next page*

*Figure 5 continued*

(**b**) $V_{1/2}$ values from *a* plotted versus holding potential. (**c–d**) Normalized *FV* relations of Shaker-iVSD and Shaker-W434F from a holding potential of 0 mV to a pre-pulse of −90 mV for variable duration (5–5000 ms) followed by a series of test pulses of −180 to +160 mV in steps of 10 mV (see inset for protocol). For Shaker-iVSD, smooth curves are fits to a Boltzmann function with $V_{1/2}$ values shown in **e** as a function of pre-pulse duration. For Shaker-W434F, the resulting distribution is a superposition of two Boltzmann curves representing the $FV_{0\ mV}$ and the mode-shifted $FV_{-90\ mV}$. Smooth curves are fits of the data to a double Boltzmann function, and the fractional amplitudes of $FV_{-90\ mV}$ component are shown in **e** as a function of pre-pulse duration. (**e**). Data points of Shaker-iVSD (red circle, right ordinate) were fitted by a single-exponential function with a time constant for entering the mode shift of 452 ms. Data points of Shaker-W434F (black circle, left ordinate) were fitted by a double-exponential function having time constants for entering the mode shift of 36 ms and 279 ms. (**f**) Comparision the time course for entering mode-shift with the development of the ionic current of Shaker-iVSD. The red data points are $V_{1/2}$ values recorded from one oocyte plotted as a function of pre-pulse duration (5–1000 ms) (see protocol in **c**), and fitted by a single-exponential function (red curve). Ionic current recorded from the same oocyte (black) was scaled and overlaid with the red curve.

conducting state only very slowly, whereas entering is significantly faster albeit slower than the gating kinetics of wild type Shaker channels.

The effects can be explained by a simple model for entering the relaxed state (*Figure 5g*), where both the *resting* and *activated* state are conducting, whereas the *relaxed* state is non-conducting. We can assign the fluorescence changes to the fast transition from *resting* to the *activated* state. The current vanishes when entering the *relaxed* state, but this transition is silent in the fluorescence changes. This model is supported by the temporal correlation between relaxation (mode-shifting) and the opening of the iVSD-channel from a holding potential of 0 mV (*Figure 5e*). Our data do not exclude the possibility that additional states exist. Channel closing and relaxation, although linked, do not necessarily have to be identical transitions, or an *activated-relaxed* state as suggested by Bezanilla and co-workers is possible (*Labro et al., 2012*; *Villalba-Galea et al., 2008*).

The permeation path is likely to be similar to those described for the ω-currents and Hv1 (*Starace and Bezanilla, 2004*; *Tombola et al., 2005*; *Takeshita et al., 2014*; *Tombola et al., 2007*; *Musset et al., 2011*). This notion is supported by the block through $Zn^{2+}$ (*Figure 3*) as the location of the involved residues are positioned at the entrance to the gating pore.

In the case of the proton and ω-currents passing through the Shaker voltage sensing domain, it is suggested that the cations follow the path normally taken up by the arginines in S4 responsible for the gating charges. In both cases, the arginines were replaced either with histidines or cysteines. In the activated wild type Shaker, these arginines are coordinated by acidic residues (*Li et al., 2014a*; *Vargas et al., 2012*; *Long et al., 2005a*); more specifically, the second, third and fourth cationic residues in S4, R3 (R368), R4 (R371) and K5 (K374), are coordinated by the acidic residues E247, E283 and E293, respectively, according to the crystal structure (*Long et al., 2007*). This coordination is likely transferred to R1 (R362) and R2 (R365) in the transition to the resting state when the S4 assumes its 'down' position (*Li et al., 2014a*; *Vargas et al., 2012*; *Starace and Bezanilla, 2004*).

In Hv1 channels in resting position, similarly, R3 (R208) is coordinated by the aspartate D112, responsible for the proton selectivity in these proton channels (*Takeshita et al., 2014*). The major difference between the voltage sensors of Shaker and Hv1 are an abundance of basic and acidic residues pointing towards the common interface, respectively. In Shaker, we can therefore assume that every acidic residue in the core of the VSD remains coordinated to an arginine whereas, in Hv1, D112 likely becomes uncoordinated in the activated position because the fifth positive charge in S4 is missing. D112 can then act as the selectivity filter facilitating proton transport (*Takeshita et al., 2014*; *Musset et al., 2011*).

In Shaker-iVSD, no residue in the core of the VSD has been mutated, indicating that the salt bridges between R1-R6 and the acidic residues of S1-S3 must have been broken by a different conformational change. At this time, it would be purely speculative to specify which salt bridge would be responsible without further information or a high-resolution structure.

We can, however, link gating of the Shaker-iVSD channel to relaxation of the VSD. Relaxation or mode shift is the displacement of the voltage dependence of charge movement and conductance to more hyperpolarized potentials upon prolonged holding at hyperpolarized potentials. Relaxation

has been linked to C-type inactivation, stabilization of the open state, length of the S3-S4 linker and N-terminus in different constructs (*Haddad and Blunck, 2011*; *Piper et al., 2003*; *Tan et al., 2012*; *Olcese et al., 1997*; *Kuzmenkin et al., 2004*; *Bruening-Wright and Larsson, 2007*; *Villalba-Galea et al., 2008*; *Priest et al., 2013*). While the S3-S4 linker and the N-terminus (T1 domain) rest intact and may still influence relaxation, in the absence of the pore, neither C-type inactivation nor open state stabilization may have an influence. In the resting and activated state, Shaker-iVSD forms a cation-selective pore whereas this pore is closed in the relaxed state, indicating that the acidic residues responsible for the cation pore are no longer accessible either by reorientation or coordination with a basic residue. Alternatively, relaxation might stabilize the 'native' conformation of the voltage sensing domain.

The conformational changes of the voltage sensing domain in the absence of the pore domain might be minor, but the development of a cation pore and the much slower kinetics in spite of the same primary sequence indicates that the stabilizing effects of the pore domain are essential for the native function. Stabilization might include direct protein-protein contacts between the voltage sensing domain and S5 (*Li-Smerin et al., 2000*), in the coupling region (*Batulan et al., 2010*; *Wall-Lacelle et al., 2011*; *Haddad and Blunck, 2011*; *Muroi et al., 2010*; *Long et al., 2005b*; *Catterall, 2010*; *Bezanilla, 2005*; *Lu et al., 2001, 2002*; *Labro et al., 2008*) or between voltage sensing domain and pore at the outer surface (*Lee et al., 2009a*; *Petitjean et al., 2015*; *Lainé et al., 2003*). Stabilization might also be evoked by the energetic interaction between the domains for instance the missing coupling or cooperativity between subunits.

In the construct used here, we removed the pore region and the C-terminus whereas the N-terminal tetramerization (T1)-domain (*Shen et al., 1993*; *Li et al., 1992*) is still present. The T1-domain is responsible for the correct assembly of Kv tetramers and can tetramerize in the isolated form (*Kreusch et al., 1998*). It is therefore possible that the Shaker-iVSD are held together in the cytosol by the T1-domain. While this leaves the possibility that the voltage sensors interact with each other, the higher sensitivity of D277H to zinc block indicates that this interaction is not responsible for the formation of the permeation pathway. The position of D277H suggests that the ions pass through the gating pore in the center of the VSD as discussed above.

The system as a whole will have a different energy landscape than the isolated domains, which becomes evident in the very different voltage ranges causing relaxation in Shaker-W434F and –iVSD. In most cases, these effects cannot be strictly separated. Although interpretation is facilitated when domains of macromolecules are considered separate entities with energetic interactions, it is essential when interpreting macromolecular structure function relations or structures of isolated domains to keep in mind that the final assembly does not necessarily behave like the sum of its components and might adopt a different conformation.

The fact that simply removing the pore domain of the Shaker $K^+$ channel leads to a voltage-gated cation channel suggests that the structural link between Kv and Hv channels might be closer than previously thought. If the phenomenon of a cation leak for the isolated voltage sensor is conserved also in other voltage-gated ion channels, it might be responsible for disease development in truncation mutants downstream of S4 (*Cox et al., 2006*).

## Materials and methods

### Molecular biology

The wild-type background Shaker H4 channel cDNA with inactivation peptide removed (IR, Δ6–46) in a pBSTA vector was used as a template for all mutants (*Batulan et al., 2010*; *Choi et al., 1991*). A cysteine was inserted into the S3-S4 linker at position A359C for simultaneous fluorescence measurements (*Mannuzzu et al., 1996*; *Cha and Bezanilla, 1997*). This mutation was present in all constructs and is not specifically mentioned. A W434F mutation was inserted in the pore region (*Perozo et al., 1993*), rendering the channel non-conducting for gating current measurements. The Shaker-iVSD construct was generated by removing all residues between I384 and the C-terminus in the Shaker-W434F-A359C background using two BglII restriction sites. The resulting C-terminus became 373... FKLSRHSKGLQIWLRYH* (S4 underlined). Site-directed mutagenesis and deletion mutagenesis were done with QuickChangeTM site-directed mutagenesis kits from Stratagene. All of the cDNA clones were sequenced to verify mutations.

## Oocyte preparation and injection

Oocytes from *Xenopus laevis* were surgically obtained. Follicular membrane was removed with collagenase type 1 A (C9891 Sigma, Oakville, Canada) in a $Ca^{2+}$-free solution (1 mg/ml). cRNA was in vitro transcribed (mMachine T7; Invitrogen), and 27–32 nl were injected into each oocyte with a concentration of 1 µg/µl using a 'nano-injector' (Drummond Scientific Co., Broomall, PA). As control, we injected oocytes with the same volume of $H_2O$. After injection, oocytes were incubated at 18°C in modified Barth solution to allow the translation, processing and embedding of the proteins into the cell membrane. The modified Barth solution contained (mM): 90 NaCl, 3 KCl, 0.41 MgSO4, 0.41 CaCl2, 0.33 Ca(NO3), 5 HEPES, 2.5 sodium pyruvate, 100 U/ml pen-strep, 5% horse serum, adjusted to pH 7.6 with NaOH. If not specified otherwise, electrophysiological recordings were performed 2–3 days after injection of cRNA. N is stated in each experiment; data were obtained from at least 3 independent oocyte preparations.

## Gene transfection and cell culture

HEK293 cells (RRID: CVCL_0045) were cultured using standard tissue culture conditions (5% CO2, 37°C) in high glucose DMEM supplemented with FBS (10%), L-glutamine (2 mM), penicillin (100 U/ml), and streptomycin (10 mg/ml) (Gibco BRL Life Technologies, Burlington, ON, Canada). Cells were generally grown to 50% confluency prior to transfection, and were transiently transfected with cDNA encoding Shaker-iVSD-A359C and a Zeocin-resistance-eGFP fusion protein by calcium phosphate precipitation. Cells were plated on glass coverslips pre-treated with poly-D-lysine 6 hr post-transfection. Whole-cell patch clamp experiments were performed 24~48 hr after transfection. Transfected cells expressing GFP were identified under a fluorescent microscope (Eclipse Ti, Nikon).

## Electrophysiology

All experiments used $Cl^-$-free solutions to minimize the contamination with endogenous oocyte $Cl^-$ currents, unless stated otherwise. All electrophysiology and fluorescence recordings were carried out at room temperature (~20°C) using the cut-open oocyte voltage-clamp, voltage-clamp fluorometry, and excised inside-out patch technique.

Cut-open oocyte voltage-clamp (*Taglialatela et al., 1992*) was performed with a CA-1B amplifier (Dagan Corp.) using GPatch software (Department of Anesthesiology, University of California, Los Angeles, Los Angeles, CA) as described earlier (*Batulan et al., 2010*; *Haddad and Blunck, 2011*). Oocytes were placed in an apparatus, comprising three electrically isolated chambers containing the external solution, and then permeabilized by adding 0.2% saponin to the external solution in the bottom chamber, giving electrical access to the cytosol. Saponin was washed out and the bottom chamber filled with internal solution. For experiments characterizing the gating current, the voltage-dependent fluorescence changes, the amplitude and the voltage dependence of ionic currents, the external solution contained (mM) 115 NMDG-MeSO$_3$, 10 HEPES, and 2 Ca(OH)$_2$ at pH 7.35, and the internal solution contained 115 NMDG-MeSO$_3$, 10 HEPES, and 2 EDTA at pH 7.35. Solutions used to characterize the permeability of Shaker-iVSD-induced ionic current to different ions (e.g., $NMDG^+$, $H^+$, $Na^+$, $K^+$, and $Cl^-$) are described in the legend of *Figure 2*. The voltage electrode was filled with 3 M KCl.

Fluorometry was performed simultaneously with cut-open oocyte voltage-clamp. For fluorescent labeling, *Xenopus oocytes* were incubated in a depolarizing solution (115 mM K-MeSO$_3$, 2 mM Ca(OH)$_2$, 10 mM HEPES, at pH 7.1) containing 5 µM tetramethylrhodamine- 5-maleimide (TMRM) at room temperature for 20 min. For fluorescence measurements, an upright fluorescence microscope (Axioskop 2FS; Zeiss) and a Photomax 200 photodetection system (Dagan Corp.) were used. Voltage-dependent fluorescence changes were measured from TMRM fluorophores attached to the cysteine at position A359C, located at the extracellular end of the S4 segment.

For whole-cell patch clamp experiments, ionic currents from transfected HEK293 cells were recorded using pCLAMP software 10.2 and an Axopatch 200B amplifier (Molecular Devices, Sunnyvale, CA, USA). Patch electrodes were fashioned from borosilicate glass (Corning 8161) with resistance about 1.5 MΩ when filled with intracellular solution. Whole-cell currents were low-pass filtered at 2 kHz, digitized at a sampling rate of 100 µs during acquisition, and stored on a microcomputer equipped with an AD converter (Axon Digidata 1322 A, Molecular Devices, Sunnyvale, CA, USA). The pipette solution contained (in mM) 140 NMDG-MeSO$_3$, 10 EDTA, 10 HEPES at pH 7.35. The

extracellular solution contained (in mM) 140 NMDG-MeSO$_3$, 1 Ca(OH)$_2$, 1 Mg(OH)$_2$, 10 HEPES, 10 Glucose at pH 7.35.

High-buffer (HB) solutions were also used for isolated characterization of Shaker-iVSD-induced currents requiring manipulation of the pH gradient. for clarity, the compositions hb solutions used to characterize the permeability of Shaker-iVSD-induced ionic current are described in the legend of *Figure 2*. for selectivity determination in oocytes, the nonspecific currents from mock (H$_2$O)-injected oocytes exposed to the same ionic conditions as the experimental group was subtracted from steady-state Shaker-iVSD-induced currents, although they were small and negligible (<2% of the total currents recorded from Shaker-iVSD-injected oocytes).

### Mode shift protocol and data analysis

Fluorescence-voltage (*FV*) relations were calculated based on the peak change in emission of the total trace. Most *FV* relationships were fit with a single Boltzmann function:

$$\frac{dF}{F} = \frac{1}{1 + \exp\left(\frac{(V_{1/2} - V)}{dV}\right)}$$

where $dF/F$ is the change in fluorescence amplitude normalized to maximal fluorescence amplitude, $V_{1/2}$ is the half-activation potential, $V$ is the membrane potential, and $dV$ is the slope factor.

For the analysis of the non-conducting Shaker-W434F mutation fluorescence, some data were best fit with the sum of two Boltzmann functions:

$$\frac{dF}{F} = a \frac{1}{1 + \exp\left(\frac{V_{1/2,1} - V}{dV_1}\right)} + (1 - a) \frac{1}{1 + \exp\left(\frac{(V_{1/2,2} - V)}{dV_2}\right)}$$

where symbols are as described above, $a$ refers to the fraction of each Boltzmann component.

The kinetics of the ionic and fluorescence traces during channel activation were fitted with a double-exponential function of the form:

$$y(t) = A_{fast} \cdot \left(1 - e^{-\frac{t}{\tau_{fast}}}\right) + A_{slow} \cdot \left(1 - e^{-\frac{t}{\tau_{slow}}}\right)$$

where $A_{fast}$ and $A_{slow}$ correspond to the amplitudes and $\tau_{fast}$ and $\tau_{slow}$ to the time constants of the fast and slow components, respectively, and $t$ is time.

To determine the speed of recovery from mode shift (relaxation), we measured fluorescence changes of Shaker-iVSD and –W434F by pulsing from 0 mV to −90 mV for various durations (5–5000 ms) before applying a series of test pulses using the cut-open oocyte voltage-clamp technique.

For Shaker-iVSD, the V$_{1/2}$ values of the *FV* curves as a function of the pre-pulse duration to −90 mV were obtained by fitting each curve to a single Boltzmann function. For Shaker-W434F, *FV* curves were fitted with double Boltzmann functions, and the resulting distribution was a superposition of two Boltzmann curves representing the *FV* relation at a holding potential of 0 mV (*FV*$_0$ $_{mV}$) and the mode-shifted *FV* relationship when pre-pulse duration at −90 mV $\geq$ 1000 ms (*FV*$_{-90}$ $_{mV}$). The fractional amplitudes of FV$_{-90}$ $_{mV}$ component are shown as a function of pre-pulse duration.

All the fluorescence, gating, and ionic current measurements were analyzed using a combination of Analysis software (Department of Anesthesiology, University of California, Los Angeles) and SigmaPlot for Windows version 12.5 (SPS, Chicago, IL, USA). Data reported throughout the text and figures are reported as mean ± SEM.

## Acknowledgements

We would like to thank Mireille Marsolais for technical support and Georges A Haddad for generating the initial construct. This work was financially supported by the Canadian Institutes for Health Research (MOP-136894 and MOP-102689) and the Natural Sciences and Engineering Research Council (DG- 327201–2012).

## Additional information

### Funding

| Funder | Grant reference number | Author |
|---|---|---|
| Canadian Institutes of Health Research | MOP-136894 | Rikard Blunck |
| Natural Sciences and Engineering Research Council of Canada | DG- 327201-2012 | Rikard Blunck |
| Canadian Institutes of Health Research | MOP-102689 | Rikard Blunck |
| Natural Sciences and Engineering Research Council of Canada | CDMC-CREATE postdoctoral fellowship | Juan Zhao |

The funders had no role in study design, data collection and interpretation, or the decision to submit the work for publication.

### Author contributions

JZ, Acquisition of data, Analysis and interpretation of data, Drafting or revising the article; RB, Conception and design, Analysis and interpretation of data, Drafting or revising the article

### Author ORCIDs

Rikard Blunck, http://orcid.org/0000-0003-4484-2907

### Ethics

This study was performed in strict accordance with the guidelines of the CDEA of Université de Montréal (licence No. 16-033).

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
