## [Decision Letter]

Thank you for submitting your article "The isolated voltage sensing domain of the Shaker potassium channel forms a voltage-gated proton channel" for consideration by *eLife*. Your article has been favorably evaluated by Gary Westbrook as the Senior Editor and three reviewers, one of whom, Kenton J Swartz (Reviewer #1), is a member of our Board of Reviewing Editors, and another is Baron Chanda (Reviewer #2).

The reviewers have discussed the reviews with one another and the Reviewing Editor has drafted this decision to help you prepare a revised submission.

Summary:

This is an interesting manuscript describing the discovery that deletion of the pore domain of the Shaker Kv channel has a profound influence on the behavior of the remaining S1-S4 voltage-sensing domains. Remarkably, this deletion causes the remaining four voltage-sensing domains to conduct cations over a wide range of voltages where fluorescence experiments show that the sensors fluctuate between resting and activated states, and that the conduction pathway only closes as the sensors enter the relaxed state that predominates at positive voltages. The permeation pathway appears to be relatively non-selective, and the authors estimate the selectivity sequence as H^+^>Na^+^=K^+^>NMDG^+^. Deletion of the pore-domain also has a profound influence on the energetics of voltage-sensor activation, causing a decrease in the valence of *FV* relations by about 5-fold. The manuscript is well-written and the data support most of the conclusions the authors advance. That the pore has such a substantial impact on the voltage sensors is really quite surprising, and we think the work is appropriate for *eLife*.

Essential revisions:

1) There is confusion about how the proton current measurements were made and what solutions were used. The authors state that they used NMDG-MES plus HEPES in most of their solutions. Is MES short for methanesulfonic acid, a commonly used anion in gating current measurements, or is it 2-(N-morpholino) ethanesulfonic acid, a common pH buffer? If it is the former, HEPES would be a good choice for neutral pH measurements, but would not be optimal for more acidic or basic pH measurements. Also, most labs studying proton currents use the appropriate buffers at high concentration. The authors should clarify and justify their choice of buffers, and it would be ideal if they could make confirmatory measurements using appropriate buffers for the different pH measurements. The authors state that the currents in Figure 1 are leak subtracted using P/4 and those in Figure 1 are unsubtracted. However, for the ion selectivity measurements in Figure 2 the legend states that currents are leak subtracted, but does not specify how. It will not be trivial to accurately quantify and subtract leak currents in those experiments and we could imagine rather large errors depending on how subtraction was done. Finally, as the conductance measured is not very proton selective, it would be more appropriate to use 'cation' channel instead of proton channel in the title and elsewhere.

2) The experiments with Zn seem quite complicated could be interpreted in multiple ways. In the Results section, the authors state: "A reduced relative fluorescence change indicates that immobilization of the voltage sensor by Zn caused the current decrease. Fluorophores attached to voltage sensors immobilized by Zn would still contribute to the total fluorescence intensity but would no longer be displaced and thus not display a voltage dependent change (dF). Accordingly, the relative fluorescence change dF/F would be lower." We would expect stabilization of a state to shift the *FV* rather than diminishing dF/F. Of course it could be that Zn produces a huge shift beyond what can be measured. But other things could be going on too. Perhaps Zn simply blocks the permeation pathway (as is invoked on the next page), and Zn binding might modify quench TMR fluorescence. The author's conclusion may be correct, but would like to hear a clearer presentation of the Zn results and what they mean. It would also be helpful if the fluorescence data in Figure 3 was presented with and without normalization so the reader could evaluate the actual data and see whether Zn only effects dF or alternatively might also be affecting F.

3) Given the data that are presented, there is a nagging concern that the expressed construct may not be the direct source of the currents measured. It would be ideal for the authors to do additional experiments to rule out the contribution of some endogenous channels. It is possible that the isolated voltage-sensing domain associates or modulates the conductance of the endogenous channel (minK like effect- see Wang and Goldstein, 1995 Neuron) rather than forming a channel on its own. The zinc experiments address those concerns to some degree but zinc is also non-specific and will inhibit many channels with exposed cysteines or histidines. We suggest that the authors attempt to express their construct in mammalian cells and also to introduce Cys residues at positions in S4 where reaction with MTS reagents could inhibit the currents measured. We are not asking for an extensive set of new experiments, but would like additional support for the main conclusion of this paper.

4) The main purpose of the paper is to investigate the VSD conformational changes associated with channel activation when the pore domain is missing. Ideally, this could be achieved by studying the behavior of the VSD in isolation. The Shaker-iVSD construct is truncated soon after the VSD, but still contains the N-terminal tetramerization domain (T1D, Li et al. Science 1992, Shen et al. Neuron 1993). While the T1D is not always necessary to induce tetramerization of the full-length channel (Tu et al. J. Biol. Chem. 1996), it forms tetramers on its own (Kreusch et al. Nature 1998), and could potentially induce VSD oligomerization in the absence of transmembrane segments S5 and S6. Knowing the oligomeric state of Shaker-iVSD can inform on whether the observed ion conduction and changes in activating transitions are the direct result of the uncoupling from the pore domain or the result of a supramolecular assembly that the VSDs are forced to adopt when the pore domain is not there. Although determining the oligomeric state of the present construct may not be straightforward, it would be good for the authors to discuss this issue.

---

## [Author Response]

[…] Essential revisions:

*1) There is confusion about how the proton current measurements were made and what solutions were used. The authors state that they used NMDG-MES plus HEPES in most of their solutions. Is MES short for methanesulfonic acid, a commonly used anion in gating current measurements, or is it 2-(N-morpholino) ethanesulfonic acid, a common pH buffer? If it is the former, HEPES would be a good choice for neutral pH measurements, but would not be optimal for more acidic or basic pH measurements. Also, most labs studying proton currents use the appropriate buffers at high concentration. The authors should clarify and justify their choice of buffers, and it would be ideal if they could make confirmatory measurements using appropriate buffers for the different pH measurements.*

MES was methanesulfonic acid, not N-morpholino-ethanesulfonic acid. We repeated the experiments at low and high pH with the appropriate buffers at higher concentration. The results did not alter significantly, and the new results are shown in Figure 2.

Since we also used N-morpholino-ethanesulfonic acid (MES) buffer now, we abbreviated methane-sulfonic acid with MeSO_3_.

*The authors state that the currents in Figure 1 are leak subtracted using P/4 and those in Figure 1 are unsubtracted. However, for the ion selectivity measurements in Figure 2 the legend states that currents are leak subtracted, but does not specify how. It will not be trivial to accurately quantify and subtract leak currents in those experiments and we could imagine rather large errors depending on how subtraction was done.*

We apologize, this was indeed ambiguously expressed. The “leak subtraction” in the case of Figure 2 was not using the P/4 protocol. We only subtracted the endogenous current from mock-injected oocytes. The endogenous current was very constant and only a small fraction of the iVSD current. The subtraction did not influence the results significantly. We replaced “leak-subtracted” with “mock-subtracted” to prevent any confusion.

*Finally, as the conductance measured is not very proton selective, it would be more appropriate to use 'cation' channel instead of proton channel in the title and elsewhere.*

We altered proton channel to cation channel throughout the manuscript.

*2) The experiments with Zn seem quite complicated could be interpreted in multiple ways. In the Results section, the authors state: "A reduced relative fluorescence change indicates that immobilization of the voltage sensor by Zn caused the current decrease. Fluorophores attached to voltage sensors immobilized by Zn would still contribute to the total fluorescence intensity but would no longer be displaced and thus not display a voltage dependent change (dF). Accordingly, the relative fluorescence change dF/F would be lower." We would expect stabilization of a state to shift the FV rather than diminishing dF/F. Of course it could be that Zn produces a huge shift beyond what can be measured. But other things could be going on too. Perhaps Zn simply blocks the permeation pathway (as is invoked on the next page), and Zn binding might modify quench TMR fluorescence. The author's conclusion may be correct, but would like to hear a clearer presentation of the Zn results and what they mean. It would also be helpful if the fluorescence data in Figure 3 was presented with and without normalization so the reader could evaluate the actual data and see whether Zn only effects dF or alternatively might also be affecting F.*

The reviewers are correct that stabilization of a state would lead to a shift in the *FV* and only if the *FV* is shifted beyond the experimental range, the dF/F would diminish. We think that this is exactly what is happening here, and we therefore called it “immobilized”.

However, the reviewers are also correct that the effects could be explained if binding of Zn to iVSD had two effects: 1) blocking on the permeation pathway and 2) removal of the quenching effect. In this case, both effects could potentially originate from different proteins as long as both effects bind Zn with the same affinity (Figure 3). Zn itself did not quench fluorescence as we did not observe a significant decrease in the total fluorescence intensity (Figure 3 top).

We added a supplementary figure (Figure 3—figure supplement 1), showing the reduction of the gating currents upon addition of Zn. If the effect of Zn on iVSD would merely be prevention of fluorescence quenching, the gating currents should not have been effected. However, the gating currents reduce like the current and the fluorescence traces indicating that indeed voltage sensor movement is impaired.

We also showed with histidine-mutants (see 3 below) that the currents originate from iVSD further supporting our interpretation.

We added the non-normalized fluorescence traces (Figure 3) and Figure 3—figure supplement 1, as well as the histidine mutants. We also altered the text to explain the interpretation more in detail.

*3) Given the data that are presented, there is a nagging concern that the expressed construct may not be the direct source of the currents measured. It would be ideal for the authors to do additional experiments to rule out the contribution of some endogenous channels. It is possible that the isolated voltage-sensing domain associates or modulates the conductance of the endogenous channel (minK like effect- see Wang and Goldstein, 1995 Neuron) rather than forming a channel on its own. The zinc experiments address those concerns to some degree but zinc is also non-specific and will inhibit many channels with exposed cysteines or histidines. We suggest that the authors attempt to express their construct in mammalian cells and also to introduce Cys residues at positions in S4 where reaction with MTS reagents could inhibit the currents measured. We are not asking for an extensive set of new experiments, but would like additional support for the main conclusion of this paper.*

We followed the advice of the reviewers and added two new supporting set of experiments to the manuscript:

As the reviewers suggested, we expressed iVSD in mammalian cells (Figure 1). The observed effects were the same, indicating that the currents do originate from iVSD itself and not endogenous currents of the oocytes.

We made a histidine mutation in iVSD mimicking the HV1 channels and significantly increased the sensitivity of the Zn block. The voltage dependence of the currents was also shifted to more hyperpolarized potentials (Figure 3). If the currents were originating from other channels, mutating iVSD would not have such a profound effect on the current. Unfortunately, the double mutant D277H-E335H did not express sufficiently.

*4) The main purpose of the paper is to investigate the VSD conformational changes associated with channel activation when the pore domain is missing. Ideally, this could be achieved by studying the behavior of the VSD in isolation. The Shaker-iVSD construct is truncated soon after the VSD, but still contains the N-terminal tetramerization domain (T1D, Li et al. Science 1992, Shen et al. Neuron 1993). While the T1D is not always necessary to induce tetramerization of the full-length channel (Tu et al. J. Biol. Chem. 1996), it forms tetramers on its own (Kreusch et al. Nature 1998), and could potentially induce VSD oligomerization in the absence of transmembrane segments S5 and S6. Knowing the oligomeric state of Shaker-iVSD can inform on whether the observed ion conduction and changes in activating transitions are the direct result of the uncoupling from the pore domain or the result of a supramolecular assembly that the VSDs are forced to adopt when the pore domain is not there. Although determining the oligomeric state of the present construct may not be straightforward, it would be good for the authors to discuss this issue.*

As suggested by the reviewers, we added a discussion of the possible role of the T1 domain to the manuscript (Discussion, ninth paragraph). We also mentioned its possible influence on development of relaxation (Discussion, seventh paragraph).

We are currently in the process of determining the oligomeric state of iVSD since it might form oligomers even in the absence of the T1 domain. This would be beyond the scope of the current manuscript, though. However, just like Hv1 channels, we do not think that an eventual oligomerization would be responsible for development of the permeation path. The D277H mutant shows that permeation occurs through the gating pore.